# HIVE: A Hyperbolic Interactive Visualization Explorer for Representation Learning

Thijmen Nijdam*    Derck Prinzhorn*    Jurgen de Heus*    Thomas Brouwer*

University of Amsterdam

{thijmen.nijdam, derck.prinzhorn, jurgen.de.heus, thomas.brouwer}@student.uva.nl

## Abstract

*We present **HIVE**, an interactive dashboard that supports exploration and interpretation of hyperbolic embeddings in deep learning. Hyperbolic spaces naturally capture hierarchical structure, yet existing visualization tools are either designed for Euclidean geometry or remain static when curvature is taken into account. HIVE closes this gap by offering 2D projections in the Poincaré disk and integrating configurable dimensionality-reduction algorithms for hyperbolic space, including CO-SNE and HoroPCA. From expert interviews, we distilled key analytic needs and realized them in four interaction modes: compare, traverse, tree, and neighbors. These modes enable real-time, multimodal analysis through semantic hierarchy tracing, geodesic interpolation, and projection comparison. A small but targeted user study demonstrates that HIVE supports practical analysis and uncovers meaningful hyperbolic structure. While currently limited to image and text embeddings, the dashboard shows promise for broader applications, such as reinforcement learning and graph discovery, highlighting HIVE's potential as a useful tool for future hyperbolic learning scenarios. Source code and a demo are available at https://github.com/thijmennijdam/HIVE.*

## 1. Introduction

Hyperbolic geometry is increasingly adopted in deep learning for modeling hierarchical, tree-like, and other relational structures that Euclidean embeddings struggle to capture [4, 17]. Because volume in hyperbolic space grows exponentially with radius, it naturally mirrors hierarchical data [12]. Empirical studies further report gains in spatial awareness, ambiguity resolution, and out-of-distribution discrimination [7]. Visualizing these embeddings is not only useful for uncovering the inner workings of hyperbolic models but also lets practitioners verify that semantic hier-

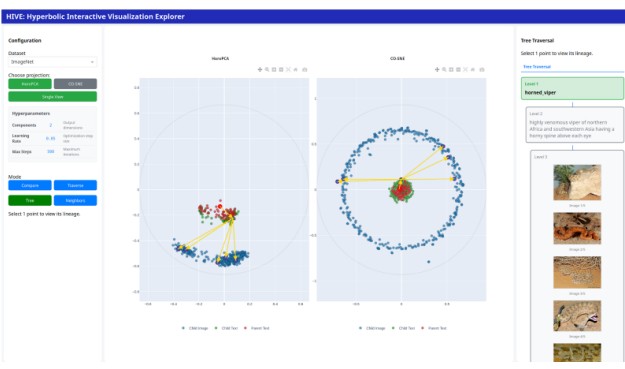

Figure 1. Our system, HIVE, an interactive dashboard for exploring hierarchical structure in high-dimensional data. Users can choose projections (HoroPCA or CO-SNE), and explore different interaction modes (Compare, Traverse, Tree, and Neighbors). The main view visualizes embeddings in hyperbolic space in the center panel, while the right panel shows detailed information for selected points.

archies and distances are preserved in the learned space.

Most existing visualization frameworks, however, are designed for Euclidean spaces and are not well-suited to represent the geometry and hierarchical structure of hyperbolic space. While visualizations compatible with hyperbolic space are mostly static and non-interactive, this limitation restricts researchers' ability to analyze and interpret hyperbolic embeddings. The following research questions are posed to assess whether a dashboard could address this gap, which are answered using an insight-based evaluation and structured survey:

- How well can an interactive dashboard support practical exploration and analysis of high-dimensional hyperbolic embeddings?
- To what extent could an interactive dashboard help users gain meaningful insights into key properties of hyperbolic learning?

To answer these questions, we conducted a requirements analysis with machine-learning researchers working on hyperbolic representations. This process revealed three main

needs: (1) interactive exploration of global hierarchical structure; (2) support for multiple projection methods; (3) inspection of individual embeddings. Based on these requirements, we built **HIVE** (Hyperbolic Interactive Visualization Explorer), a dashboard that renders 2D projections in the Poincaré disk using two reduction algorithms, CO-SNE [5] and HoroPCA [1], and offers four interaction modes: *compare*, *traverse*, *tree*, and *neighbors*. These allow users to explore both the global and local structure of hyperbolic spaces flexibly and intuitively. Figure 1 shows the interface in tree mode and dual view, displaying the embeddings of both projection methods side by side. Our contributions are:

- An interactive system for visualizing hyperbolic embeddings with multiple projection options and four modes of interaction: *compare*, *traverse*, *tree*, and *neighbors*.
- An evaluation that combines insight-based analysis and a structured user study.
- A modular open-source framework supporting dataset switching, real-time updates, and rich user interaction.

## 2. Related work

Our work intersects hyperbolic representation learning and interactive visual analytics, connecting advances in hyperbolic embeddings with the human-centered tools needed to interpret them.

**Hyperbolic representation learning**  Nickel and Kiela's [14] Poincaré embeddings first showed that the exponential volume growth of hyperbolic space is ideal for encoding taxonomies and other hierarchies. A follow-up Lorentz formulation retained this curvature-aware bias while improving optimisation efficiency [15]. Since then, hyperbolic embeddings have advanced image classification and uncertainty calibration [9], zero-shot recognition [11], and vision-language retrieval. Examples include MERU [4], HyCoCLIP [17], and HySAC for safe content moderation [18]. These successes increase the demand for tools that reveal the latent hierarchical structure of hyperbolic spaces and allow researchers to verify model behavior. Addressing this need is the primary goal behind the development of HIVE.

**Interactive visual analytics for embeddings**  Multimedia analytics combines visualization, human interaction, and analytical routines to explore complex model representations [24]. The MM4AI agenda frames such tools as a means for human-AI teaming [8, 20, 23]. Concrete systems include ReVACNN, which lets users steer CNN training in real time [2], and a dashboard for probing transformer attention [10].

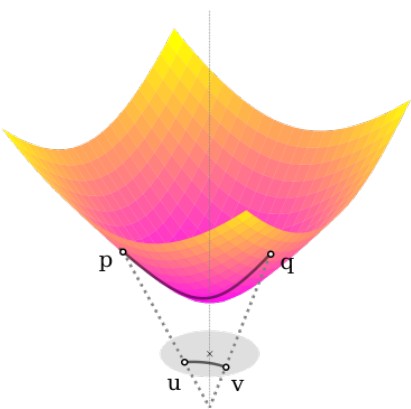

Figure 2. Two common models of hyperbolic space [15]. Top: the Lorentz (hyperboloid) model represents points on the upper sheet of a two-sheeted hyperboloid. Bottom: the Poincaré disk model maps the same geometry inside the unit disk. Both models describe identical hyperbolic structures but use different coordinate systems and reveal different visual properties.

Generic embedding viewers, such as Embedding Projector [21], Embedding Atlas [19], and WizMap [22], support scalable navigation but assume Euclidean geometry. Conversely, browser-based hyperbolic graph viewers render geodesic layouts yet target only graph topology and offer limited interaction [13]. None of these tools provides interactive, multimodal exploration of high-dimensional hyperbolic embeddings.

HIVE fills this gap by combining projection methods for hyperbolic space with four dedicated interaction modes that let users analyze both global hierarchies and local neighborhoods in real time.

## 3. Background

This section outlines the key concepts that underlie our system. First, we review hyperbolic geometry. Next, we discuss two projection methods that form the basis of our visualization approach.

### 3.1. Hyperbolic space

The Poincaré ball is the most convenient model for visualization. It represents $d$-dimensional hyperbolic space as the open unit ball in $\mathbb{R}^d$:

$$\mathbb{D}^d = \{\, p \in \mathbb{R}^d : p_1^2 + \cdots + p_d^2 < 1 \,\}.$$

In this model, the geodesics (shortest paths) are circular arcs that meet the boundary orthogonally. Although distances, areas, and volumes are distorted relative to Euclidean space,

the model is conformal, so angles are preserved. This property makes it well-suited to visualizing hierarchical structure.

The hyperboloid, or Lorentz, model embeds $d$-dimensional hyperbolic space in $\mathbb{R}^{d+1}$ as:

$$\mathbb{H}_d = \{\, p \in \mathbb{R}^{d+1} : p_0^2 - (p_1^2 + \cdots + p_d^2) = 1,\ p_0 > 0 \,\},$$

With geometry defined by the Lorentz product:

$$p \circ q = p_0 q_0 - (p_1 q_1 + \cdots + p_d q_d).$$

Because isometries can be expressed linearly, distances and geodesics have simple closed forms, giving the model good numerical stability for optimization [15]. The hyperboloid can be projected to the Poincaré ball using stereographic projection, recovering conformality.

### 3.2. Hyperbolic projection methods

HoroPCA extends principal component analysis from Euclidean to hyperbolic space [1]. Standard PCA relies on linear projections and therefore ignores negative curvature. HoroPCA instead projects data onto horospheres, surfaces orthogonal to a point at infinity within the Poincaré ball, naturally preserving hyperbolic geometry. Many hyperbolic models output embeddings in the Lorentz representation, so we first map them to the Poincaré ball; this conversion preserves geometry while improving interpretability in two dimensions.

CO-SNE adapts the t-SNE algorithm to hyperbolic space [5]. Where t-SNE minimizes the Kullback–Leibler divergence between pairwise similarities in Euclidean space, CO-SNE measures similarity with hyperbolic distance inside the Poincaré ball, thereby maintaining the hierarchical relationships encoded by curvature. As with HoroPCA, embeddings are transferred to the Poincaré ball before optimization.

## 4. Methodology

This section outlines the methodological components of HIVE, the system architecture, and the interaction modes that enable detailed inspection of the embedding space.

### 4.1. Requirements Analysis

We first conducted a requirements analysis with researchers in the field to develop a tool supporting meaningful exploration of hyperbolic representations. The identified core requirements led directly to the following key features: (1) interactive exploration of the two-dimensional Poincaré disk, enabling complementary views of hyperbolic geometry; (2) configurable projection methods, such as CO-SNE and HoroPCA, facilitating structural comparisons between embedding techniques; and (3) capabilities for selecting single or multiple embeddings for detailed analysis. Additionally,

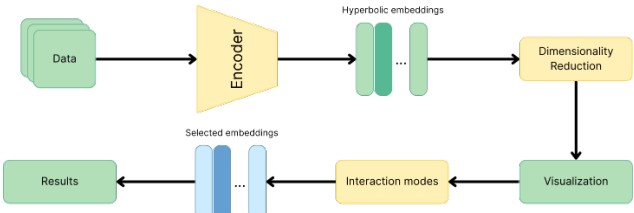

Figure 3. HIVE processing pipeline. Samples are embedded in hyperbolic space, reduced to two dimensions, and rendered for interactive exploration.

researchers emphasized the utility of a traversal tool that allows users to view the intermediate embeddings between two selected projected points, assisting in comprehending transitions within hyperbolic embedding spaces. These insights significantly guided the design and implementation of HIVE.

### 4.2. System Architecture

HIVE in its current form visualizes two hierarchical multimodal datasets: GRIT [6] and ImageNet-1K [3]. Each image and text sample is embedded with a configurable encoder; the current implementation uses HyCoCLIP [17].

The encoder produces a high-dimensional representation in the Lorentz model, which is converted to the equivalent Poincaré-ball form. HIVE then applies two curvature-aware reduction techniques, HoroPCA and CO-SNE, mapping the high-dimensional points to two-dimensional coordinates inside the Poincaré disk. These 2D projections form the basis for all interactive views. Figure 3 shows the full HIVE processing pipeline.

### 4.3. Dashboard Design

The dashboard consists of three panels, as shown in Figure 1: a configuration panel on the left, a central visualization area, and a detail panel on the right. In the configuration panel, users can select the dataset, projection method, interaction mode, and optionally enable Dual View. All selections made in this panel directly determine what is shown in the central visualization area. When Dual View is enabled, the central panel displays two projection methods side by side. These views are fully linked as clicking a point in one panel automatically highlights the corresponding point in the other, enabling direct visual comparison. The content of the detail panel varies depending on the selected interaction mode.

### 4.4. Interaction Modes

The four interaction modes: *compare*, *traverse*, *tree*, and *neighbors* allow distinct ways for users to explore and analyze the embedding space. Each mode reconfigures the dashboard to support a specific analytical intent.

**Compare** Users can simultaneously visualize and analyze up to five selected embeddings, each shown with its associated image or textual description. This allows direct exploration and comparison of the embedding space, as shown in Figure 4a.

**Neighbors** Users select a point to inspect its $k$ nearest neighbors. Given a dataset $\mathcal{D} = \{x_1, \ldots, x_N\}$, a query point $x$, and an arbitrary distance function $d$, the nearest neighbors are defined as:

$$\text{NN}_k(x) = \arg\text{top}_k \{d(x, x_i) \mid x_i \in \mathcal{D}\},$$

where $\arg\text{top}_k$ selects the $k$ points with the smallest distances to $x$. The distance function $d$ can be Euclidean, hyperbolic, or another metric. The rightmost panel highlights the images or texts of these neighbors, enabling users to assess spatial and semantic proximity. An example of a point and its corresponding neighbors is shown in Figure 4b.

**Tree** Users visualize hierarchical relationships around a selected embedding, showing parent and child connections in the semantic taxonomy. In this mode, the right panel renders the local hierarchy, displaying both textual and visual context for the selected point, its parent, and its children. The tree visualization makes explicit the relationships between abstract and concrete concepts, where nodes closer to the origin should represent higher-level semantics. Figure 4c shows an example hierarchy for a class in ImageNet.

**Traversal** Users can create paths between two embeddings in Lorentz space, interpolating intermediate points along the shortest geodesic. Given two points $s, t \in \mathbb{L}^n$, intermediate points are computed using logarithmic mapping at the origin, following formulas presented in [17]. After exponential mapping back to hyperbolic space, each interpolated point is matched to its nearest neighbor using the same nearest-neighbor retrieval method defined above. Duplicate matches are removed, resulting in a discrete approximation of the geodesic path. The length of the path is configurable through the left panel. The detail panel visualizes the sampled sequence, with each intermediate embedding linked to its original sample and metadata. Figure 4d demonstrates an example traversal with a path of length 3.

## 5. Evaluation

Evaluating multimedia analytics systems requires balancing qualitative insight with quantitative assessment. As argued by North [16], insight-based evaluation captures rich, open-ended user interactions, while benchmark-driven assessments provide structured metrics. Building on this principle, we adopt an evaluation approach consisting of two

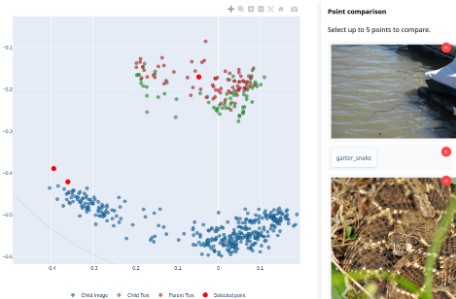

(a) **Compare** mode displays selected embeddings along with their associated samples for direct inspection.

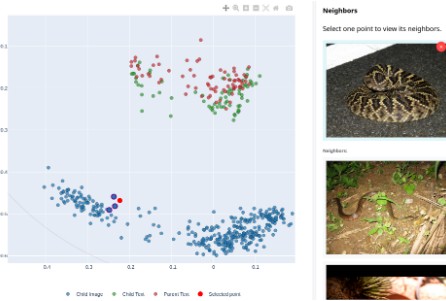

(b) **Neighbors** mode highlights the $k$ nearest neighbors of a selected point based on hyperbolic distance.

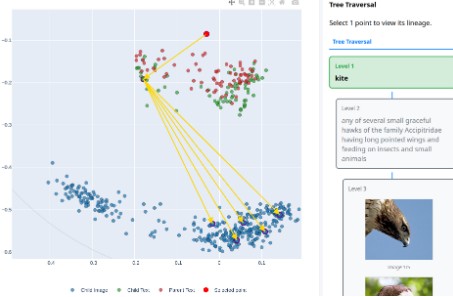

(c) **Tree** mode visualizes local hierarchical structure around a selected embedding, showing parent and child relationships.

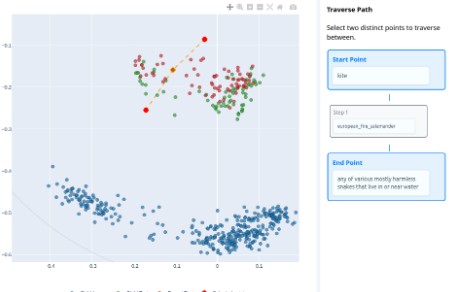

(d) **Traverse** mode visualizes a geodesic path between two selected embeddings. Intermediate points are sampled along the geodesic and matched to their nearest neighbors in the dataset.

Figure 4. The four interaction modes in HIVE, each combining a central visualization panel with a context-sensitive detail panel.

Table 1. Average Likert-scale scores for each evaluation aspect. Scores range from 1 (strongly disagree) to 5 (strongly agree).

| Aspect | Mean Score |
| --- | --- |
| Usefulness | 4.5 |
| Quality of Visualization | 4.8 |
| User Experience | 4.7 |
| Overall Average | 4.7 |

complementary components: (1) an insight-based evaluation to explore user interaction and reflection, and (2) a structured Likert-scale survey to quantify perceptions of usability. Due to the specialized nature of hyperbolic learning, we limited our evaluation to a small but targeted group of five expert users with relevant domain experience.

## 5.1. Likert-Scale Survey

We conducted a structured Likert-scale survey to quantify user perceptions of Usefulness, Quality of Visualization, and User Experience. The full survey is provided in the supplementary material. Each dimension was measured using multiple items on a five-point Likert scale (1 = Strongly Disagree, 5 = Strongly Agree). Results are summarized in Table 1.

The survey indicates that participants evaluated the dashboard very positively across all aspects. Quality of Visualization received the highest score (4.8), highlighting the clarity and expressiveness of the visual components. User Experience (4.7) suggests the interface was intuitive and responsive, while Usefulness (4.5) reflects the perceived value for exploring hyperbolic embeddings.

Participants were also invited to provide open-ended feedback. Visualization of the tree structure was particularly appreciated, and users found the traversal feature valuable for revealing hierarchical relationships.

## 5.2. Insight-Based Evaluation

An insight-based evaluation was performed with the same participants. After a brief introduction to the dashboard, participants explored the system freely without predefined tasks. They then reflected on what they learned about the embedding space and provided informal feedback, which also inspired ideas for further use cases.

A key shared observation was the difference between projection methods: participants noted that CO-SNE often produced more coherent neighborhoods than HoroPCA. Another consistent finding was the global spatial arrangement of modalities as text embeddings appeared near the center of the Poincaré disk, while image embeddings were positioned near the boundary. This indicates that the HyCo-Clip model [17] successfully captured the greater generality of text compared to the specificity of images.

Both the tree and traversal features were received positively. Tree mode helped validate semantic hierarchies and quickly reveal non-hierarchical structures in the embeddings, while traversal mode supported the inspection of intermediate representations. In some cases, participants noted semantic inconsistencies in the resulting trees or traversals, reinforcing the dashboard's value as a diagnostic tool for identifying problematic embeddings.

## 5.3. Suggested Improvements

Evaluation feedback highlighted several directions for future development. Participants consistently recommended additional features and dataset support to enhance the dashboard's utility for research.

Suggestions included visualizing entailment cones to better reveal local structure in hyperbolic space, as this is more relevant than standard neighbor retrieval in non-Euclidean geometry. Another frequently requested improvement was support for loading and visualizing custom or larger datasets. Larger datasets would also enhance the usefulness of traversal mode, which relies on a higher density of embeddings to generate meaningful paths between points.

## 6. Conclusion

This work addressed the lack of dedicated tools for exploring and interpreting hyperbolic embeddings. By conducting a requirements analysis with researchers, we identified the need for interactive, multimodal visualization tools that surpass the limitations of static hyperbolic plots and Euclidean-focused dashboards. This motivated the development of HIVE, a modular dashboard that enables the analysis of high-dimensional hyperbolic embeddings through various projection methods and interaction modes. HIVE supports both global and local structure analysis with its comparison, traversal, tree, and neighbors functionalities.

To evaluate HIVE, we formulated two research questions. The first, concerning the effectiveness of an interactive dashboard for practical exploration and analysis of high-dimensional hyperbolic embeddings, was assessed through a structured Likert-scale survey. The results indicated high ratings for usefulness, visualization quality, and user experience, suggesting that the tool is effective and intuitive for practical analysis.

The second research question focused on the extent to which the dashboard facilitates meaningful insights into the properties of hyperbolic learning. An insight-based evaluation showed that participants were able to interpret semantic structures in the embedding space, such as the global arrangement of modalities and the hierarchical relationships between general and specific embeddings. These findings suggest that HIVE enables users to uncover important aspects of hyperbolic learning.

While current insights are closely linked to the dashboard's main use case, visualizing semantic structure in multimodal hyperbolic embeddings, participants also identified several promising directions for future applications. These suggestions highlight HIVE's broader potential for supporting a variety of research scenarios. To realize this potential, further extensions are needed to accommodate new datasets and analytical tasks. In summary, our evaluation on a small but targeted group of experts suggests that HIVE provides a robust foundation for the exploration of hyperbolic embeddings and offers a valuable starting point for future developments in this area.

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
