# OpenReview forum: "HIVE: A Hyperbolic Interactive Visualization Explorer for Representation Learning"
_thecvf.com/ICCV/2025/Workshop/BEW — BEW 2025 Oral_

### Official Review · Reviewer_f6uP · 2025-07-06
**In summary, while this paper does not have any major technical contributions (it uses existing methods in its entire pipeline), it is an extremely useful research tool and should see adoption in multiple research areas such as out-of-domain robustness, concept-based learning (concept bottleneck models, for instance, frequently use hierarchical vocabularies), and discovering biases in datasets and models. This is quite an useful tool for data-centric ML, and I am convinced that as interest in hyperbolic spaces grows in the community over the years, there will be increased adoption of this tool.**

**Rating:** 5
**Confidence:** 3

**Review:**

This paper proposes a visualization and retrieval tool for hyperbolic embedding spaces. Hyperbolic representations are useful in that their inductive bias relies on capturing natural hierarchies in the data. For example, ImageNet labels are derived from WordNet synsets, which describe a hierarchical relationship amongst objects in the wild.

HIVE uses the HyCoCLIP encoder to generate the embeddings, perform dimensionality reduction with hyperbolic-friendly extensions of standard reduction algorithms such as PCA, t-SNE, that allows users to visualise the data in 2-D, hyperbolic space. In addition to this, there is a human-in-the-loop feature that allows users to analyze embeddings further, using distance based retrieval techniques.

Here are certain points that I wish were covered in the paper:

- There are connections to adaptive model diagnosis that HIVE can enable. See [1] for inspiration. It would be nice if the authors can provide some perspective for how HIVE can be leveraged this way.

- Another point that I wish was covered in the paper is the connection to knowledge graph embeddings. Knowledge graphs are also naturally hierarchical, and there exist a number of approaches using hyperbolic representations in this domain. See [2] and [3], for example. In fact, I am curious what the authors think of using the embeddings in [2] in the HIVE framework.

I have two points of concern:

- It may be a slight misunderstanding, but the examples shown in Figure 5 - are they extracting the training data that represents each embedding, or is it more a qualitative analysis? Because training data extraction is a significant privacy risk, and ideally one shouldn't be using HIVE to extract training data from frozen encoders.

- Scalability - How efficient are the HoroPCA and Co-SNE operations? This seems to be the bottleneck for the whole framework.

In summary, while this paper does not have any major technical contributions (it uses existing methods in its entire pipeline), it is an extremely useful research tool and should see adoption in multiple research areas such as out-of-domain robustness, concept-based learning (concept bottleneck models, for instance, frequently use hierarchical vocabularies), and discovering biases in datasets and models. This is quite an useful tool for data-centric ML, and I am convinced that as interest in hyperbolic spaces grows in the community over the years, there will be increased adoption of this tool.

[1] Gao, Irena, et al. "Adaptive testing of computer vision models." Proceedings of the IEEE/CVF International Conference on Computer Vision. 2023.

[2] https://hazyresearch.stanford.edu/hyperE/

[3] Chami, Ines, et al. "Low-dimensional hyperbolic knowledge graph embeddings." arXiv preprint arXiv:2005.00545 (2020).

---

### Official Review · Reviewer_QT1T · 2025-07-07
**HIVE is timely needed tool to advance hyperbolic DL, and my decision is a weak accept**

**Rating:** 4
**Confidence:** 5

**Review:**

**Summary**

The paper presents HIVE, an interactive visualization dashboard specifically designed for analyzing hyperbolic embeddings in DL. Conventional visualization tools are predominantly tailored to Euclidean geometry, which limits their capacity to represent the inherent hierarchical and curved nature of hyperbolic spaces. HIVE addresses this gap by enabling real-time, dynamic exploration of high-dimensional hyperbolic embeddings through interpretable 2D projections. Particularly notable are its user-driven interaction modes: comparison, tree, traversal, and neighbors. They facilitate structure analysis in an intuitive manner. The modular architecture further enhances its utility, making it adaptable across diverse datasets and analytical tasks. The deliberate integration of two theoretically grounded dimensionality reduction methods, e.g., CO-SNE and HoroPCA, demonstrates a careful consideration of the geometric properties of hyperbolic space, contributing to both the methodological rigor and practical value of the proposed system. With minor revisions, this work will likely become a foundational work for interactive visualization research of hyperbolic embeddings.


**Justification**

With the growing adoption of hyperbolic embeddings in LVM settings, there is a clear need for visualization tools beyond Euclidean constraints.
HIVE fills this methodological gap, enhancing interpretability grounded in the geometry of hyperbolic space. The paper is logically structured. While the paper makes a compelling contribution to the field, a few areas of improvement could be addressed in the camera-ready version to further strengthen the paper.

1. **Elaborate more extensively on each interaction mode by including richer visual illustrations or extended examples.** Given the unique contribution of these modes to enabling intuitive exploration of hyperbolic embeddings, a more comprehensive description with concrete use cases would improve clarity and reader engagement. It would also be advisable to utilize the full page limit to convey the technical and practical insights each mode affords.

2. **The lack of quantitative benchmarks remains a weakness.** The inclusion of meaningful metrics would help to objectively substantiate the interpretability claims made for the system.

---

### Official Review · Reviewer_B7SZ · 2025-07-07
**Potentially useful visualization dashboard, limited technical contribution and insights**

**Rating:** 2
**Confidence:** 4

**Review:**

**Summary:**
The paper proposes a dashboard to visualize existing hyperbolic visualization methods like HORO-PCA, and Co-SNE, while providing interactive functionalities to  explore the hyperbolic embedding space through four main modes: Compare, Traverse, Tree, and Neighbors.

**Strength:**
- Interactive visualization can be a good educational and probing tool to understand the properties of the learned representation space. Hence, the proposed dashboard seems useful for the hyperbolic problem domain.

- The four functionalities provided seem reasonable to understand the sample’s hierarchical relationship in the embedding space.

- The proposed dashboard’s utility has somewhat been somewhat validated by the survey conducted on the domain experts.


**Weakness:**

- The paper presents limited research novelty. It primarily integrates existing visualization techniques into a dashboard, focusing on UI features rather than introducing new algorithms or theoretical insights. While the interactive functionalities are useful, the contribution is largely in interface design, with minimal technical or engineering innovation.


- The sample size for the survey of 4 candidates is very small and can be highly biased. If the paper wants to show user study it requires a larger sample size than 4.

- Current visualization is limited to Imagenet and GRIT datasets, the paper doesn’t provide analysis on other hierarchical datasets to generalizability/applicability of the proposed dashboard.

- Paper could have provided more visualization of each of the functionalities. In its current form, the figures are less informative and are difficult to understand. More details and examples would help understand the dashboard application better.

- The Supplement was unintuitive. There was no readme, I tried to run the scripts using commands.sh, but there was no UI script, all I could achieve was plots saved to some specific directories. An additional video of UI working, if not a properly documented repository would have made it easier to visually check the UI.

- It would be beneficial to discuss whether there are alternative, more objective methods of evaluating the dashboard’s effectiveness beyond user feedback, which can be subjective.


**Overall,** I see the need for a UI dashboard like HIVE, and the tool could be useful for practitioners working with hyperbolic embeddings. However, in its current form, the paper focuses more on UI development than on significant research contributions. It would benefit from further development, broader evaluation, and more detailed discussion in several areas (see weaknesses above). My current recommendation is Borderline Reject. If the authors address these issues and the AC(s) decide(s) to conditionally accept the paper, I would have no objection to that.

---

### Decision · Program_Chairs · 2025-07-09

**Decision:**

Accept (Oral)

**Comment:**

The majority of the reviews agree towards accepting the paper.  The authors should do their best to address the comments of the reviewers in their final version.